# Dual-Task Gait Stability after Concussion and Subsequent Injury: An Exploratory Investigation

**DOI:** 10.3390/s20216297

**Published:** 2020-11-05

**Authors:** David R. Howell, Scott Bonnette, Jed A. Diekfuss, Dustin R. Grooms, Gregory D. Myer, Julie C. Wilson, William P. Meehan

**Affiliations:** 1Sports Medicine Center, Children’s Hospital Colorado, Aurora, CO 80045, USA; julie.wilson@childrenscolorado.org; 2Department of Orthopedics, University of Colorado School of Medicine, Aurora, CO 80045, USA; 3The Micheli Center for Sports Injury Prevention, Waltham, MA 02453, USA; Greg.Myer@cchmc.org (G.D.M.); william.meehan@childrens.harvard.edu (W.P.M.III); 4The SPORT Center, Division of Sports Medicine, Cincinnati Children’s Hospital Medical Center, Cincinnati, OH 45219, USA; Scott.Bonnette@cchmc.org (S.B.); Jed.Diekfuss@cchmc.org (J.A.D.); 5Ohio Musculoskeletal & Neurological Institute, Ohio University, Grover Center, Athens, OH 45701, USA; groomsd@ohio.edu; 6Division of Physical Therapy, School of Rehabilitation and Communication Sciences, College of Health Sciences and Professions, Ohio University, Grover Center, Athens, OH 45701, USA; 7Division of Athletic Training, School of Applied Health Sciences and Wellness, College of Health Sciences and Professions, Ohio University, Grover Center, Athens, OH 45701, USA; 8Departments of Pediatrics and Orthopaedic Surgery, College of Medicine, University of Cincinnati, Cincinnati, OH 45219, USA; 9Division of Pediatrics, University of Colorado School of Medicine, Aurora, CO 80045, USA; 10Division of Sports Medicine, Boston Children’s Hospital, Boston, MA 02115, USA; 11Departments of Pediatrics and Orthopaedic Surgery, Harvard Medical School, Boston, MA 02115, USA

**Keywords:** mild traumatic brain injury, adolescent, pediatric, locomotion, postural stability, inertial measurement units, accelerometers

## Abstract

Persistent gait alterations can occur after concussion and may underlie future musculoskeletal injury risk. We compared dual-task gait stability measures among adolescents who did/did not sustain a subsequent injury post-concussion, and uninjured controls. Forty-seven athletes completed a dual-task gait evaluation. One year later, they reported sport-related injuries and sport participation volumes. There were three groups: concussion participants who sustained a sport-related injury (n = 8; age =15.4 ± 3.5 years; 63% female), concussion participants who did not sustain a sport-related injury (n = 24; 14.0 ± 2.6 years; 46% female), and controls (n = 15; 14.2 ± 1.9 years; 53% female). Using cross-recurrence quantification, we quantified dual-task gait stability using diagonal line length, trapping time, percent determinism, and laminarity. The three groups reported similar levels of sports participation (11.8 ± 5.8 vs. 8.6 ± 4.4 vs. 10.9 ± 4.3 hours/week; *p* = 0.37). The concussion/subsequent injury group walked slower (0.76 ± 0.14 vs. 0.65 ± 0.13 m/s; *p* = 0.008) and demonstrated higher diagonal line length (0.67 ± 0.08 vs. 0.58 ± 0.05; *p* = 0.02) and trapping time (5.3 ± 1.5 vs. 3.8 ± 0.6; *p* = 0.006) than uninjured controls. Dual-task diagonal line length (hazard ratio =1.95, 95% CI = 1.05–3.60), trapping time (hazard ratio = 1.66, 95% CI = 1.09–2.52), and walking speed (hazard ratio = 0.01, 95% CI = 0.00–0.51) were associated with subsequent injury. Dual-task gait stability measures can identify altered movement that persists despite clinical concussion recovery and is associated with future injury risk.

## 1. Introduction

Advancements with return-to-play guidelines have led to improved patient care for individuals who sustain a concussion [1]. However, objective elements pertaining to diagnosis, treatment, and recovery monitoring remain challenging. Common clinical concussion assessments, such as symptom inventories or computerized neurocognitive tests, do not possess an adequate level of reliability or construct validity [2] and may not be able to detect subtle physiological impairments that persist after clinical recovery [3]. Thus, further work is required to improve concussion assessment approaches. Gait instability has been identified as a post-concussion deficit that lingers despite symptom, neurocognitive, and neurophysiologic recovery among adolescent athletes [4,5,6].

Although the addition of a symptom-free waiting period prior to return to play has reduced rates of repeat concussion for athletes [1], there has been increasing recognition of other negative outcomes following return to play after concussion, such as an increased risk of a subsequent musculoskeletal injury [7]. Existing meta-analyses report a two-fold increase in odds of sustaining a sport-related injury in the year after a concussion compared to non-concussed athletes [8,9]. While increased injury risk has been observed across youth, collegiate, and elite athletes, few studies have examined prognostic measures for future injury risk stratification upon return-to-play after a concussion. One study observed common clinical concussion assessments do not provide predictive information regarding subsequent musculoskeletal injury risk [10], while poor dual-task gait (i.e., completing a cognitive task during gait) is associated with injuries sustained during the year after a concussion [11,12]. While both studies suggest that dual-task approaches can be one approach to identifying post-concussion injury risk, they did not include a non-concussed control group to serve as a reference for typical performance of similar athletes. Given that both attention [13] and motor control [14] deficits are associated with higher injury risk independent of a concussion history, dual-task gait performance may provide increased sensitivity to quantify post-concussion deficits associated with future injury risk.

While average walking speed under dual-task conditions identifies post-concussion recovery, such linear- and temporal-based metrics do not expose the effects of concussion on underlying gait structure, precluding insight into critical movement patterns more sensitive to detecting deficits in movement stability [4,15]. Specifically, nonlinear- and spatial-based approaches that quantify the structure of stride-to-stride fluctuations expressed during gait may help explain the sensorimotor deficits underlying slower walking speeds after concussion that remain, despite clinical recovery [15,16]. One nonlinear technique, cross recurrence quantification analysis (CRQA), has recently demonstrated that the structure of gait variability differs between adolescents with a recent concussion and uninjured adolescents, providing novel insight into impaired movement beyond traditional metrics [15,17]. However, these studies were limited by cross-sectional designs of concussion and uninjured controls. Deviations from the structure of optimal, healthy gait variability may indicate that movement is either unstable or overly stable. Specifically in terms of the structure of gait variability, overly stable behavior may indicate a difficulty transitioning between motor behaviors (e.g., pivoting to avoid a collision during an athletic competition) and unstable behavior may indicate an inability to maintain motor behavior in the face of external perturbations (e.g., maintaining upright stance during a collision) [15,17,18].

CRQA has been used to identify metrics associated with reduced gait stability following concussion. Further work is needed to identify if unnoticed dynamic motor control deficits under dual-task conditions persist after clinical recovery. This novel approach may provide information regarding injury risk in the year following post-concussion return-to-play clearance, connecting recent epidemiology data [8] to a behavioral metric [15]. Therefore, the purpose of our investigation was to examine linear and nonlinear dual-task gait variables obtained after post-concussion return-to-play clearance among adolescent athletes who did and did not sustain an injury within the subsequent year, and among uninjured control participants. We hypothesized that participants who reported sustaining a sport-related injury in the year following their concussion would demonstrate altered CRQA measurements relative to individuals who sustained a concussion and reported no injury in the subsequent year and uninjured controls who reported no injury in the year after assessment.

## 2. Materials and Methods

### 2.1. Study Design and Participants

We conducted a prospective cohort study of youth athletes who sustained a concussion and uninjured healthy control participants. We assessed concussion participants after their treating physician cleared them to return to pre-injury levels of sports participation (i.e., concussion recovery based on standard clinical guidelines). For the purposes of this investigation, we used gait data collected during an in-person assessment, several days after return-to-play clearance for the concussion group. No injuries were reported between the return-to-play clearance physician visit and study assessment. Approximately one year after the in-person assessment, we sent a follow-up questionnaire to all participants regarding injuries they sustained during the year.

Participants with a concussion were treated within a single sports medicine clinic associated with a tertiary care regional children’s hospital. Physicians providing care were board-certified sports medicine physicians, who diagnosed the injury consistent with the definition provided by international consensus [19] and made return-to-play decisions independent of the study. No patients were hospitalized as a result of the concussion. All concussions occurred during sport or sport-like activities (e.g., falling from ground level). Control participants were athletes who presented to an injury prevention center for an injury prevention evaluation. We excluded those younger than 8 years of age or older than 20 years of age, those who reported a concussion in the year preceding the evaluation (not counting the concussion for which they were seen within this investigation), those whose gait data were not obtained at the study visit, those whose injuries involved a high-velocity impact (e.g., motor vehicle collision), those with a co-existing lower extremity injury at the time of concussion, or those with a history of permanent memory loss, learning disability, Down syndrome, or developmental disability. The study protocol was reviewed and approved by the institutional review board. All participants, and parents/guardians if under the age of 18, provided written informed consent to participate in the study.

We conducted an in-person gait evaluation that was conducted when participants were cleared to return to sports after their concussion. Then, approximately one year after this in-person gait evaluation, we sent participants a follow-up questionnaire to document any new injuries sustained. Within the questionnaire, participants reported the time spent participating in sports over the preceding year, and if they sustained a sport-related injury. Consistent with prior studies, we asked those who reported an injury to describe the body region, specific diagnosis, and the date of the injury [11,20]. A previous study among community level Australian football players suggests that approximately 80% of athletes could correctly identify the number of injuries and body regions affected within a 12 month period, using an injury history recall questionnaire [21]. A limitation to this approach, however, is that this previous work also indicates that only 61% of athletes correctly recalled the number of injuries, regions, and diagnoses sustained in the previous year [21].

For the purposes of this study, we defined a subsequent injury as an acute, sport-related injury diagnosed by a healthcare professional that resulted in time missed in sport participation. We did not include repetitive stress or developmental injuries (e.g., Osgood-Schlatter disease, shin splints) as a subsequent injury within our analyses. Based on whether they were tested after a concussion or as a healthy control, plus the response to the questionnaire, participants were categorized into three possible groups. First, those with a concussion who reported sustaining a subsequent injury in the year after their concussion (CONC-INJ). Second, those with a concussion who did not report any subsequent injuries during the year after their concussion (CONC-UNINJ). Third, controls who were healthy at the time of testing and did not sustain any other injury in the year after in-person testing (CTRL-UNINJ).

### 2.2. Clinical Evaluation

During the in-person evaluation, participants completed the Post-Concussion Symptom Scale (PCSS) [22], where they rated each concussion symptom from 0 (asymptomatic) to 6 (maximum severity). We calculated the sum of responses to provide an overall symptom severity score. As all participants had been given clearance to return to sports, their symptoms were near zero. However, given the lag time (approximately 12 days) between return-to-play clearance and assessment, as well as the non-specific nature of the PCSS, some participants may have reported symptoms at the time of assessment not necessarily indicative of concussion, while others may have reported symptoms that were related to the concussion.

Participants also completed the dual-task tandem gait test, consistent with previously described studies and standardized instructions [23,24]. Patients were instructed to walk heel-to-toe without shoes along a strip of tape 3 m in length, make a 180 degree turn beyond the end of the tape, and then return to their starting point with the same heel-to-toe gait as quickly as possible. Times were recorded to the nearest hundredth of a second using a stopwatch or smartphone. During the three trials, participants completed one of three different cognitive tasks, while simultaneously walking heel-to-toe: serial subtraction by 6 s or 7 s from a randomly presented 2 digit number, reciting months in reverse order starting from a randomly selected month, or spelling a five-letter word backward. Each participant completed one type of task during the duration of each trial, selected randomly by the test administrator. Using this approach, previous work has identified a high test–retest reliability [25].

### 2.3. Instrumented Dual-Task Gait Evaluation

In addition to tandem gait, participants walked in a typical and self-selected pattern, while wearing a set of inertial sensors (Opal Sensor, APDM Inc., Portland, OR, USA) placed along the lumbar spine at the lumbosacral junction and on the dorsum of each foot with an elastic belt [26,27]. As with prior dual-task CRQA investigations [15], participants were asked to walk and simultaneously complete a similar mental task to that described as a part of the tandem gait protocol above. A similar set of tasks were used (subtraction, spelling backwards, months in reverse order), but the same specific elements were not repeated during the assessment. Data were obtained at a sampling frequency of 128 Hz. For these devices, the frequency band was 2.40–2.48 GHz ISM band, the calibrated frequency ranged from 150 kHz to 80 MHz, and the synchronization between devices was ≤1 ms. The accelerometers had a range of +/−2 g, a bandwidth of 50 Hz, and a resolution of 14 bits. Using Mobility Lab 2.0 [26,27], we first calculated average walking speed, averaged across each of the five dual-task trials completed. Average walking speed provides an independent and complementary measure to the CRQA variables, given that CRQA provides an indication of the underlying gait variability organization regardless of how fast a person walks [15]. While average walking speed provides an over-arching view of walking behavior, CRQA provides an index of the coordinative patterns that form the stride-to-stride variations in gait.

### 2.4. CRQA Data Processing and Analysis

We calculated four CRQA variables: percent determinism, average diagonal line length, laminarity, and trapping time (see below for a description and interpretation of each). The calculations [28,29] and methods have been described in depth previously [15]. In brief, this technique is an extension of recurrence quantification analysis [30] that we apply to index dynamic gait characteristics of time series obtained from the inertial sensors placed on the feet [31,32,33]. Specifically, in the current analysis, we analyzed the vertical acceleration time series (i.e., the z axis when the accelerometers are fixed with the x and y axes facing forward and left, respectively) of each foot. To perform CRQA, we used a variable radius value to ensure each recurrence plot maintained a fixed recurrence rate of 5.0% [34,35]. We utilized individually determined delays and embedding dimensions for each participant due to individual stride length and timing variations [15]. Using the mutual information approach, a delay for each participant was chosen [36]. Likewise, false nearest-neighbor analysis was performed to select an embedding dimension [37] for each participant. On average, a delay of 20.94 and an embedding dimension of 5.00 were selected for the reconstructed state space. The maximum normalized distance was used to rescale the distance matrix (see [28]). The minimum line length for vertical and diagonal lines was set to 2 consecutive points. We performed CRQA using custom-written MATLAB (The MathWorks, Inc, Natick, MA, USA) scripts and open-source CRQA toolbox functions [38,39]. Other commonly reported CRQA variables (i.e., percent recurrence, maximum diagonal line length, maximum vertical line length, and Shannon entropy) were not reported due to the selection of a variable radius and the variable trial length, which affects the outcome of each of these measures [15].

Percent determinism indicates the proportion of recurrent points that form diagonal line sequences—with higher percent determinism indicating higher predictability in the coupling of the two signals (i.e., foot accelerometer profiles) [29]. Similar to percent determinism, a longer average diagonal line length indicates the two signals spent more time in the same regions of reconstructed phase space and is an indication of stronger coupling between time series [29]. Laminarity is similar to percent determinism, but instead, it applies to the percentage of recurrent points that make up vertical lines. Whereas diagonal lines are related to the level of coordination (coupling) between two signals [29], vertical lines indicate how often the signals exhibit laminar behavior—a type of behavior where the signals become “stuck” exhibiting the same behavior [34,40]. Likewise, trapping time is similar to average diagonal line length, but it applies average length of recurrent points that make up vertical lines. Trapping time indicates the average amount of time of laminar behavior (i.e., “stuck” or unchanging gait).

### 2.5. Statistical Analysis

Continuous variables are presented as means (standard deviation); categorical variables are presented as the number included within the group and corresponding percentage. We first compared demographic, injury, and clinical characteristics between the three groups (concussion + subsequent injury; concussion, no subsequent injury; control) using one-way analyses of variance (ANOVA) and Fisher’s exact tests. We then compared linear gait measures, nonlinear (CRQA) gait measures, and clinical measures between the three groups using univariable ANOVAs. If an omnibus test was statistically significant (*p* < 0.05), we conducted Tukey post hoc tests to determine between-group differences. To examine if the obtained measurements were associated with the hazard of injury in the subsequent year following assessment, we constructed a series of univariable Cox proportional hazards models. The outcome variable was time to injury in the year subsequent to the assessment, and the predictor variable was dual-task steady-state gait variables for linear (average gait speed) and nonlinear (CRQA) measurements. Statistical significance was set at α = 0.05, and all tests were two-sided. Statistical analyses were performed using Stata version 15 (StataCorp, College Station, TX, USA).

## 3. Results

We enrolled a total of n =111 participants (n = 71 concussion, n = 40 control). Of those, n = 48 (45%) completed the one-year follow-up questionnaire. We excluded one concussion participant, as gait performance data were not available. Thus, we included a total of N = 47 participants in our analysis. There were no significant age (14.3 ± 2.6 vs. 15.1 ± 2.8 years; *p* = 0.14), sex (53% female vs. 50% female; *p* > 0.99), concussion history (45% vs. 41%; *p* = 0.70), concussion symptom resolution time (37 ± 39 vs. 38 ± 43 days; *p* = 0.95), or initial symptom severity (20 ± 19 vs. 23 ± 17; *p* = 0.55) differences between those who were included and excluded in our analysis. Of the included participants, n = 32 were assessed after post-concussion return-to-play clearance, and n = 15 were uninjured control participants. Based on their subsequent injury status, we divided the participants into three groups: concussion participants who reported a subsequent injury in the year after their concussion (n = 8: CONC-INJ), concussion participants who did not report a subsequent injury in the year after their concussion (n = 24; CONC-UNINJ), and uninjured control participants (n = 15; CTRL-UNINJ).

Demographic characteristics were similar between the three participant groups (Table 1). Concussion participants were assessed an average of 12 ± 9 days after receiving clearance to participate in unrestricted athletic activities. Both CONC-INJ and CONC-UNINJ groups had a significantly greater proportion of participants with a history of concussion and a significantly higher initial symptom severity than the CTRL-UNINJ group (Table 1). On average, participants completed the follow-up injury questionnaire approximately one year after their final assessment (Table 2). The majority (n = 5) of the subsequent injuries were acute orthopedic, lower extremity injuries, while n = 3 were concussions (Table 2). No subsequent injury required hospitalization. There were no significant differences between the groups in the number of reported hours/week they spent participating in organized sports, or in the number of sport seasons they participated in during the year following assessment (Table 2).

The CTRL-UNINJ group demonstrated significantly faster dual-task walking speed than the CONC-INJ group (Table 3; mean difference = 0.19 m/s; 95% CI = 0.07, 0.32). No significant differences were observed between CONC-INJ and CONC-UNINJ groups, or between the CONC-UNINJ and CTRL-UNINJ groups. No significant differences were observed between the three groups for dual-task tandem gait time or symptom severity (Table 3).

The CONC-INJ group demonstrated significantly higher diagonal line length (Figure 1B; Cohen’s d = 1.27) and trapping time (Figure 1D; Cohen’s d = 1.54) than the CTRL-UNINJ group, while the CONC-UNINJ group was not significantly different than the CTRL-UNINJ group on these measures. The CONC-INJ group demonstrated significantly greater percent determinism (Figure 1A) and laminarity (Figure 1C) than the CTRL-UNINJ group. Similarly, the CONC-UNINJ group demonstrated significantly greater percent determinism (Figure 1A) and laminarity (Figure 1C) than the CTRL-UNINJ group. Dual-task diagonal line length, trapping time, and average walking speed were significantly associated with time-to-injury in the year after the assessment (Table 4).

## 4. Discussion

The results of our exploratory investigation indicate three dual-task gait performance measures—diagonal line length, trapping time, and average walking speed—were associated with post-concussion sports injury incidence during the year after return-to-sport clearance. The current results, although observed in a small sample of participants, extend prior observations suggesting that the dual-task gait metrics may be a useful addition for clinicians who care for athletes with concussion to identify risk of subsequent injury upon clinical recovery [11,12].

Dual-task gait speed is a measure reflecting the ability to complete a motor and cognitive task simultaneously following concussion [4,41]. Observations to date indicate post-concussion dual-task gait impairments can persist even after athletes are cleared to return to sports [42]. Thus, it has been theorized that continued dual-task dysfunction exists after return-to-play clearance and may contribute to subsequent injury [7], further supported by data from recent investigations [11,12] along with our current investigation. However, rather than measures that index the fine scale aspects of gait dynamics (i.e., the structure of gait variability), these studies used a gross measure of gait—specifically the average speed of gait—as the outcome measure. Measures that index the structure of gait variability (i.e., CRQA) may reflect persistent impairments that underlie reduced dual-task gait speed and/or provide novel therapeutic targets to potentially restore the neuromuscular control abilities that support normal, healthy gait variability.

Dual-task diagonal line length and trapping time measures obtained via CRQA may provide useful insights regarding future injury risk and potential treatment recommendations. Higher diagonal line length values are reflective of a stronger coupling in the time series signal of the left and right feet within a gait cycle [29]. Similarly, trapping time indicates the amount of time spent during consecutive gait cycles, where the signals are unchanging or “stuck” [15]. These diagonal line measures examine the predictability of coupling between signals obtained on each foot. Percent determinism and laminarity are simply the percentage of recurrent points that, respectively, form the diagonal and vertical lines within the cross recurrence plot (see [15]). The two measures provide a gross indicator of how often the two signals (i.e., the left and right foot accelerations) are exhibiting similar behavior across time. On the other hand, diagonal line length and trapping time are measures that index how long, on average, the similar behavior is exhibited by two signals once the recurrent behavior begins. Deviations from “normal” (in our study: uninjured controls) in a greater or lesser direction may indicate an unhealthy movement stability state under dual-task conditions, which reflect a more realistic sport-like demand than more common single-task concussion assessments (e.g., static postural control). We observed higher values for both diagonal line length and trapping time for the CONC-INJ compared to the CTRL-UNINJ group, and no significant differences between the CONC-UNINJ and CTRL-UNINJ, suggesting an overly stable or “stuck” movement pattern that could inform injury risk profiles after concussion. We recognize, however, that our small sample size and the presence of outliers within this analysis (Figure 1) may have influenced the observed significant differences, although the large effect sizes suggest some clinical significance. Gait pattern behavior via CRQA may reflect an altered movement pattern that has not fully recovered after a concussion and cannot be detected by traditional clinical methods. Thus, the elevated diagonal line length and trapping time (overly stable) response to a cognitive perturbation during gait despite being returned to play may be one contributing factor to the observation of increased post-concussion injuries [8].

Given that subsequent post-concussion sport-related injuries are likely multifactorial, identification of a multimodal set of prognostic variables are needed to further advance clinical practice. The observed CRQA measures are associated with the hazard of injury in the year after concussion, aligning with existing theories [18,43,44] suggesting that deviations from the structure of normal, healthy movement variability (i.e., control participants) is a key factor in understanding post-concussion subsequent injury risk. As with past research [17], our results suggest analyses that quantify the structure of gait variability provide complementary information to traditional linear metrics and a combination of both types of analyses have been found to best classify athletes with a concussion from uninjured athletes. In one review paper, the authors proposed that primary motor cortex dysfunction after a concussion may result in delayed peripheral muscle integration, thereby reducing neuromechanical responsiveness and increasing injury risk [45]. Others have theorized perception–action coupling deficits after concussion that lead to improper temporal movement execution or incorrect body positioning during sports, thereby increasing injury risk [46]. Our work provides experimental data supporting these theories, specifically that movement dysregulation persists following clinically observed concussion recovery. Measurement via CRQA may therefore provide a method to identify one potential contributing factor to the increased injury risk in the subsequent year. Furthermore, in addition to being a possible diagnostic tool for predicting future injury following a concussion, the dependent measures of the current study may also provide objective methods to evaluate the effects of training protocols on damaged motor control systems following injuries such as concussion. Training protocols have been shown to be effective in modifying posture, gait, and/or trunk sway after other neurologically relevant events, such as stroke [47,48], bilateral vestibular loss [49,50], Parkinson’s disease [51,52], and total hip arthroplasty [53,54].Similar training methods may be considered in future studies investigating athletes with a concussion.

### Limitations

Our study was limited in several ways, and our findings should be interpreted accordingly. Our subsequent injury data were obtained via self-recall and assessed injury status and average time spent playing sports over the course of an entire year. Therefore, subsequent injury data may have been inaccurate and susceptible to recall bias. Our participant sample was also comprised of patients who were seen at a specialty care concussion clinic. The sample may have sustained more severe injuries and possess different characteristics than those seen in other healthcare settings, so our findings are not generalizable to other populations. We did not conduct an a priori power analysis. Thus, our small sample size and large amount of missing data from participants who did not complete the one-year follow-up questionnaire limited our ability to draw generalizable conclusions, and future follow-up studies with larger cohorts and prospective surveillance are needed to confirm the results we observed.

## 5. Conclusions

Our results indicate gait behavior measured by two CRQA measures (dual-task diagonal line length, dual-task trapping time) are associated with post-concussion injuries sustained during sports in the year after clearance to resume sport participation. Clinicians should consider incorporating analyses that quantify the structure of gait variability exhibited during dual-task movement assessments as a method to identify movement disruptions and potential future injury risk during concussion return-to-play decisions, although further studies confirming these findings are required before widespread clinical implementation.

## Figures and Tables

**Figure 1 sensors-20-06297-f001:**
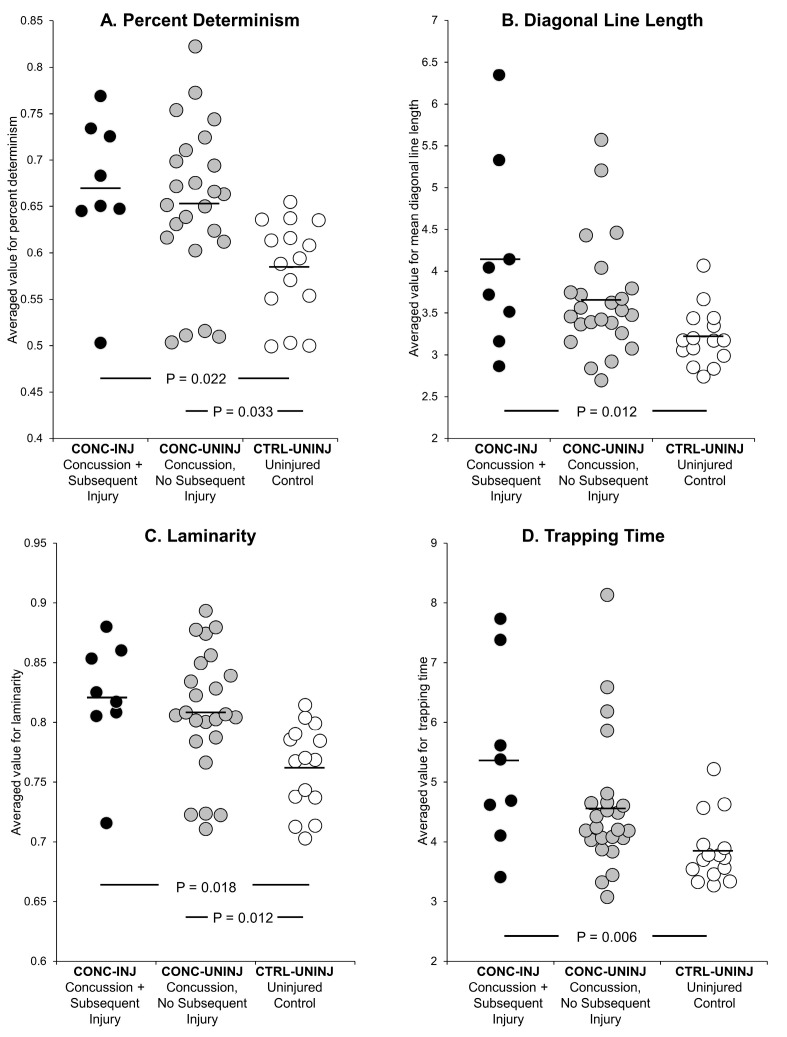
Distribution of individual data points for each of the nonlinear (CRQA) measurements obtained during the assessment after return-to-play clearance. The black bars represent the mean value in each group.

**Table 1 sensors-20-06297-t001:** Demographic, injury, and clinical characteristics of the three participant groups.

Variable	CONC-INJ Concussion + Subsequent Injury (n = 8)	CONC-UNINJ Concussion, No Subsequent Injury (n = 24)	CTRL-UNINJ Control, No Subsequent Injury (n = 15)	*p* Value
**Age (years)**	15.4 (3.5)	14.0 (2.6)	14.2 (1.9)	0.40
**Sex (female)**	5 (63%)	13 (46%)	7 (53%)	0.86
**Assessment Time** **(days post-injury †)**	50.8 (57.5)	45.7 (21.4)	28.6 (21.6)	0.18
**Assessment Time** **(days after return-to-play clearance)**	9.1 (9.0)	12.9 (9.8)	-	0.31
**Symptom Resolution Time (days post-injury)**	41.7 (56.8)	32.7 (19.9)	-	0.50
**Height (cm)**	163.1 (13.3)	160.8 (13.5)	158.2 (11.7)	0.66
**Mass (kg)**	63.3 (16.5)	54.3 (15.6)	48.3 (12.0)	0.12
**LOC at Time of Injury**	1 (13%)	2 (8%)	-	>0.99
**History of Concussion**	5 (63%)	14 (58%)	2 (13%)	0.02 *
**Initial Symptom Severity (PCSS score)**	19.6 (12.7)	31.2 (18.1)	2.4 (3.6)	<0.001 *

† Days post-injury for the control group indicates the number of days the assessment occurred after their initial evaluation. * Pairwise follow-up comparisons indicated both concussion groups had a significantly greater proportion of participants with a history of concussion and a significantly higher initial symptom severity than the control group.

**Table 2 sensors-20-06297-t002:** One year follow-up information and injury characteristics for participants who sustained a concussion, returned to sports, and sustained a subsequent injury.

Variable	CONC-INJ Concussion + Subsequent Injury	CONC-UNINJ Concussion, No Subsequent Injury	CTRL-UNINJ Control, No Subsequent Injury	*p* Value
**Hours of Week in Organized Sport Participation During the Year After Assessment**	11.8 (5.8)	8.6 (4.4)	10.9 (4.3)	0.37
**Number of Sport Seasons Completed During the Year after Assessment**	2.7 (1.5)	3.0 (0.9)	3.7 (0.5)	0.09
**Follow-up Time** **(days from assessment—questionnaire completion)**	369 (21)	374 (13)	377 (12)	0.41
**Type of Subsequent Injury**	Lower extremity injury: 5 Ankle sprain: 2 Ankle fracture: 1 Hamstring strain: 1 Knee sprain: 1 Concussion: 3	-	-	-
**Days Missed Due to Subsequent Injury**	45 (69)	-	-	-
**Time from Concussion to Subsequent Injury**	158 (91)	-	-	-

**Table 3 sensors-20-06297-t003:** Clinical and linear dual-task gait measures obtained during the assessment after return-to-play clearance.

Variable	CONC-INJ Concussion + Subsequent Injury	CONC-UNINJ Concussion, No Subsequent Injury	CTRL-UNINJ Control, No Subsequent Injury	*p* Value
**Dual-task self-selected Average Walking Speed (m/s) ***	0.76 (0.14)	0.84 (0.15)	0.95 (0.14)	0.009
**Dual-task Cognitive Accuracy (% correct)**	88.6 (11.2)%	89.2 (15.4)%	95.5 (4.5)%	0.24
**Dual-task Tandem Gait Time (s)**	19.2 (5.4)	17.6 (5.7)	16.1 (5.1)	0.78
**Symptom Severity (PCSS score)**	2.0 (4.9)	3.1 (6.3)	1.0 (1.9)	0.44

* *p* < 0.05.

**Table 4 sensors-20-06297-t004:** Cox proportional hazards results, describing the association between dual-task gait measures and time to subsequent injury.

Predictor Variable	Hazard Ratio	Standard Error	95% Confidence Interval	*p* Value
**Percent Determinism**	1.92	0.86	0.80, 4.63	0.15
**Diagonal Line Length ***	1.95	0.61	1.05, 3.60	0.03
**Laminarity**	3.24	2.44	0.74, 14.22	0.12
**Trapping Time ***	1.66	0.36	1.09, 2.52	0.02
**Average Walking Speed ***	0.01	0.02	0.00, 0.51	0.02

* Significantly associated with time to injury in the year after the clinical/gait assessment.

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
