# Peer review of "Dual-Task Gait Stability after Concussion and Subsequent Injury: An Exploratory Investigation"

_sensors, 2020, doi:10.3390/s20216297_

Round 1
Reviewer 1 Report
The author proposed dual-task gait adaptability measures among adolescents who did/did not sustain a subsequent injury post-concussion, and uninjured controls. The results are useful. The problem of the manuscript is that it needs an entire figure to illustrate the research. In addition, some results need to be present in more figures. The author can refer to "The Effect of Treadmill Walking on Gait and Upper Trunk through Linear and Nonlinear Analysis Methods" for this point. Although the author analyzed the results, there is no equation to introduce the proposed mothed. It is difficult for the reader to reuse it.
Reviewer 2 Report
Interesting analysis. Perhaps I am missing some of this, but I would recommend that authors clarify the following to improve the readability and message, especially in the abstract and methods sections:
Study sample: Were any hospitalized patients included, or were all concussed patients from mild sports concussions? Did any of them have positive CTs? The 3 groups should be CLEARLY defined in the methods.
What constituted a "subsequent injury"? an injured ankle, for instance, will affect gait quite differently than a subsequent concussion. If subsequent brain injury - how severe was it? If subsequent peripheral body injury - how severe was it? Were these treated in hospital/underwent surgery? These questions should be clarified. The authors may be limited by the group sizes of 8, 24, and 15, however heterogeneous injuries should really not be grouped together.
At what point was gait assessed? Was it uniformly at 12 months after the initial concussion, and were there any considerations on when the subsequent injury was? For instance, a subsequent injury 3 days before the gait analysis would obviously affect the measure more than a subsequent injury 300 days before the gait analysis.
Reviewer 3 Report
Q1: What is the novelty compared to references [11] and [12]
Q2: What is the novelty compared to the author's previous work (e.g., [15])
Q3: The terminology should be improved, when referring to gait stability, adaptability, and movement responsiveness. The authors use all three (almost interchangeably), saying that the results indicate a change in those. However, they all are different concepts and the authors should clarify which one of them is most appropriate in their analysis and results, and provide the clear definition. It is also beneficial if they state in the introduction what exactly the CRQA can evaluate among those three when applied to gait analysis.
Q4: What do the result say in terms of dual-task assessment? It is not clear how the cognitive task influences the motor task. Also, shouldn't these dual-task results be accompanied with respective single-task assessment? To demonstrate the necessity of dual-task evaluation, you should
also compare these results with the outcomes of single task assessments.
Q5: Although the authors say method and calculation is described in previous work, there should still be some indication on what are the time series evaluated during gait. The results cannot be fully interpreted if a description of what is measured both in the motor and cognitive aspect is not provided. For instance, line 295, what are the signals obtained on each foot?
Q6: The results support the hypothesis of altered CRQA measures, but more physical interpretation should be provided. For instance, while some generic interpretation has been given to the four CRQA variables, it should be given a more specific and biomechanical interpretation for the current results, referring to the specific measurements and data collected in the current results. For instance, what What are the neuro-mechanical interpretation on the 4 metrics?
Q7: Some interpretation on why Diagonal line length and trapping time are indicators of future injuries, while percent determinism and laminarity aren't should be provided.
Q8: How do you discuss the CRQA results in relation to the linear metrics?
Q9: You also mention that the metrics are related to movement coordination, can you please specify which metric is indicative of coordination and why? and what do the results say about coordination?
Q10: Why more stuck means overly stable (line 307)? has this been proven before? Otherwise, please remove or refine with more details, since gait stability has many different meanings
Reviewer 4 Report
The authors present an exploratory study using two dual-task movement assessments for patients who suffered concussive sports injuries. This is an area of relevance and of high interest to several members in the sports medicine and traumatic brain injury field and can be directly used to diagnose and predict prognosis in patients. It also provides clinicians with additional information pertaining to the extent of the injury, enabling them to design better rehabilitative regimens on a case-by-case basis.
The applicability of these methods for head injuries that are unrelated to sports activities (domestic injuries and vehicular accidents for instance) is also evident.
The study design, as claimed by the authors, is preliminary but does provide a rather insightful picture of how the proposed methods can be of great importance. The limitations of the study have been clearly laid out and conclusions are well founded and within reason with the data collected.
I recommend the publication of this manuscript in Sensors in its current form.
Reviewer 5 Report
To the authors: I commend the authors for undertaking a difficult study design with a critical research question. Strengths of this paper include identifying a critical research question with implications for concussion management, using nonlinear analyses to assess gait, and the use of a figure with individual data-points. I feel that there are several significant weaknesses to the design and methodology which limit the impact of the work, however. First, the sample size is underpowered, with only 8 participants with concussion who report injury and 24 who did not report injury. The statistical power of this study is therefore limited and differences found between groups cannot be interpreted with confidence. Second, the use of a self-report survey for injury counts over the course of a year introduces doubt as to the accuracy and completeness of data. There is also a large amount (over 50%) of missing follow-up data. The combination of these three factors severely limits my confidence regarding conclusions derived from this data, unfortunately.
Introduction
General- solid development of rationale, identifying the gap in the literature and how this study would address the need
L82-83: the word “surveillance” is not appropriate, given that the injuries were not recorded by a medical team but rather self-report from the participants. Please change to …who sustained a concussion who did not self-report an injury in the following year…” or something similar.
Materials and Methods
L133: Using the cited study to support using a follow-up questionnaire is a bit misrepresented. While the point you make is accurate, the authors go on to say that the 80% you mention does not include diagnosis, and that only 61% could recall exact number of injuries, region, and diagnoses. The “take home message” from that study also indicates: “Self-reported injury history data for sports injury research cannot be relied upon with confidence.”
L129: While this statement is true, you cannot say that any symptoms at the time or assessment are separate from concussion, either, without pre-injury information. Please add a clarification.
CRQA data: is there any available reliability data for these outcomes (i.e., determinism, diagonal leg length, laminarity and trapping time)? Or is it inherently unreliable given the nonlinear approach to analysis? If the latter point is the case, maybe within-subject change is a more appropriate way to analyze this kind of data than group-level comparisons.
L191: Since ANOVA is being used, were the data assessed for normality prior to analysis?
L201: Were any a priori power analyses conducted? If not, were any/could any post-hoc power analyses be provided? With a sample size this small, and the CONC-INJ group being <10 participants, this would be a critical piece of information.
Results:
L208: The large amount (62% of controls, 55% concussed) of missing data for follow-up is troubling for interpreting these outcomes. This should be added to the limitations section.
General: I applaud the authors for using the distribution plot to show individual data points, but I feel that the numerical descriptive data for these outcomes should be provided in the Results, as well.
Discussion
General: You touch on this to some degree, but I am struggling with how these measures may have any clinical impact, outside of their potential to predict further injury. For example, can these outcomes be “trained” or “improved”? If so, would that theoretically reduce injury risk?
L326: You need to add a follow-on sentence as to why you are disclosing the self-recall as a limitation. Recall bias is a significant concern, especially given the year-long follow-up period and the fact that these were mostly adolescents.
Figure 1
General: Are these differences considered “large” or clinically meaningful?
General: There appear to be significant outliers for diagonal line length and trapping time, at least. Can you comment in the discussion on how this may have influenced the significant differences, especially given the small sample size?
Round 2
Reviewer 3 Report
Authors addressed all comments.
Reviewer 5 Report
The authors have addressed my concerns.